# Strong Interference Elimination in Seismic Data Using Multivariate Variational Mode Extraction

**DOI:** 10.3390/s24227399

**Published:** 2024-11-20

**Authors:** Zhichao Yu, Yuyang Tan, Yiran Lv

**Affiliations:** 1Key Laboratory of Deep Petroleum Intelligent Exploration and Development, Institute of Geology and Geophysics, Chinese Academy of Sciences, Beijing 100029, China; yuzhichao@mail.iggcas.ac.cn (Z.Y.); ylpk8@mail.iggcas.ac.cn (Y.L.); 2Frontiers Science Center for Deep Ocean Multispheres and Earth System, Key Lab of Submarine Geosciences and Prospecting Techniques MOE, College of Marine Geosciences, Ocean University of China, Qingdao 266100, China

**Keywords:** mechanical vibration, interference elimination, seismic data, multivariate variational mode extraction

## Abstract

Seismic data acquired in the presence of mechanical vibrations or power facilities may be contaminated by strong interferences, significantly decreasing the data signal-to-noise ratio (S/N). Conventional methods, such as the notch filter and time-frequency transform method, are usually inadequate for suppressing non-stationary interference noises, and may distort effective signals if overprocessing. In this study, we propose a method for eliminating mechanical vibration interferences in seismic data. In our method, we extended the variational mode extraction (VME) technique to a multivariate form, called multivariate variational mode extraction (MVME), for synchronous analysis of multitrace seismic data. The interference frequencies are determined via synchrosqueezing-based time-frequency analysis of process recordings; their corresponding modes are extracted and removed from seismic data using MVME with optimal balancing factors. We used synthetic data to investigate the effectiveness of the method and the influence of tuning parameters on processing results, and then applied the method to field datasets. The results have demonstrated that, compared with the conventional methods, the proposed method could effectively suppress the mechanical vibration interferences, improve the S/Ns and enhance polarization analysis of seismic signals.

## 1. Introduction

Seismic technologies, both active and passive, are highly effective tools for reservoir characterization and evaluation. Active source techniques such as time-lapse seismic (4D seismic) are employed to detect fluid migration due to production processes over time [1,2,3]. Furthermore, microseismic monitoring is utilized to assess hydraulic fracturing stimulation, coal mining, CO_2_ injection and storage, and associated seismic hazards [4,5,6,7]. Additionally, the ambient-noise seismic interferometry method is used to complement traditional seismic reflection methods in petroleum exploration and development, helping to identify and monitor hydrocarbon reservoirs [8,9], as well as to image subsurface structures, particularly in areas where active seismic sources are not practical. Continuous monitoring data are typically recorded using three-component geophones deployed either in boreholes or on the surface, depending on the specific application [10]. During both temporary and permanent seismic acquisition, it is crucial to minimize interferences by placing the seismic sensors at a sufficient distance from known noise sources while maximizing the detection of seismic/microseismic events. This involves a careful analysis of the expected noise sources and their patterns, as well as the probable locations of seismic activity. Nevertheless, long-duration high-energy interferences produced by the mechanical vibrations (e.g., fracturing pump and working oil pumps) or power facilities in the seismic monitoring area may be unavoidably recorded [1,11]. These interferences have the potential to seriously deteriorate the quality of the recorded data and make it difficult to process seismic signals. Therefore, reliable techniques must be developed to extract and remove strong interferences from seismic data.

Numerous methods have been developed to improve the quality of seismic data by reducing or eliminating noise. Noise can generally be categorized into two types: random noise and strong interference noise. Commonly used methods for suppressing noise include frequency-domain filters (e.g., high-pass filter and notch filter), morphological component analysis [12], iterative trimmed and truncated mean filter method [13], randomized principal component analysis [14], time-frequency analysis methods (e.g., wavelet transform and synchrosqueezing transform method) [15,16,17], and sparse representation-based de-noising methods [18,19]. Most of these methods are primarily aimed at reducing random noise or harmonic interferences. However, conventional methods commonly struggle to eliminate the non-stationary interferences while preserving effective seismic signals, resulting in the loss of important information related to the subsequent seismic phase identification and arrival picking. Advanced methods such as unsupervised deep learning, including convolutional neural networks (CNNs) and variational autoencoders, leverage large datasets to efficiently learn the noise patterns and suppress them [20,21]. These methods require high-quality labeled data for training and the availability of various seismic monitoring scenarios needs to be considered. Unsupervised learning methods do not require labeled data, making them especially useful for situations where obtaining labeled data is challenging or impractical [22]. This flexibility allows them to be applied across a wide range of seismic monitoring scenarios, enhancing the ability to identify and mitigate noise without the need for extensive labeled datasets.

Mode decomposition-based algorithms, such as empirical mode decomposition (EMD) and variational mode decomposition (VMD), have been developed to decompose complex signals into several intrinsic mode functions (IMFs) with different characteristic time scales [23,24,25,26,27]. In recent years, these methods and their variants have been utilized to analyze and de-noise seismic and microseismic data [28,29,30,31,32]. In the EMD, the original signal is decomposed into a set of IMFs by iteratively extracting the high-frequency components and their associated envelopes. These IMFs, which correspond to the target signal as determined via signal feature analysis, are then reconstructed to obtain the de-noised signal. Similarly, the VMD method decomposes the signal into multiple modes based on an optimization framework [23]. Additionally, multivariate extension of mode decomposition-based algorithms (i.e., multivariate EMD, multivariate VMD) have been developed for processing multivariate data (e.g., multichannel signals), resulting in IMFs with aligned frequency ranges [24,26]. Recently, variational mode extraction (VME) has been proposed as a special form of VMD that extracts only one specific mode at a time and is used in various fields, including biomedical signal processing and fault diagnosis [33,34,35,36,37].

The VME method is a powerful tool for extracting band-limited components by isolating a specific mode with a predetermined approximate center frequency from a univariate input signal. However, the effectiveness of VME depends on selecting appropriate parameters, such as the center frequency and bandwidth of the modes to be extracted [35,37]. In practice, strong vibration interferences often contaminate multicomponent or multitrace seismic data on the receivers adjacent to the interference source. Ensuring a common frequency component among all channels of input data is crucial for seismic data de-noising and analysis. Therefore, we propose the use of a multivariate extension of VME (MVME) to address the problem of mode alignment when applying VME to multivariate input data. In this paper, we developed an interference noise elimination method for seismic data by using multivariate VME (MVME). The proposed method achieves strong interference removal from multichannel seismic signals by identifying the frequencies of the interference components and then extracting the interference signals. It employs MVME to extract and separate individual modes from raw seismic records with the approximate center frequencies and optimal penalty parameters. In the following sections, we first introduced the principles of VME, multivariate extension of VME and the workflow of the proposed method. Then, we systematically evaluated the performance of the proposed method using synthetic data. Finally, we used two field data examples to demonstrate the effectiveness of the proposed method in enhancing the S/Ns and facilitating subsequent data processing (e.g., seismic signal polarization analysis).

## 2. Methods

### 2.1. Variational Mode Extraction (VME)

Variational mode decomposition (VMD) is an advanced signal processing approach that has gained popularity in recent years for its ability to decompose non-stationary signals into a set of quasi-orthogonal intrinsic mode functions (IMFs) [25]. Unlike empirical mode decomposition (EMD) approaches, VMD addresses issues of mode mixing and non-optimal reconstruction by iteratively searching for modes that minimize a variational constraint [25,26]. The constrained variational problem can be defined as follows:(1)$$\begin{array}{l} \underset{\left\{u_{k}(t)\},\{\omega_{k}\right\}}{\min}\left\{\sum_{k=1}^{K}{\left\|\partial_{t}\left[\left(\delta (t)+\frac{j}{\pi t}\right)*u_{k}(t)\right]e^{-j\omega_{k}t}\right\|}_{2}^{2}\right\}\\ \begin{array}{ll} s.t. & \sum_{k=1}^{K}u_{k}(t)=x(t) \end{array} \end{array},$$
where *K* represents the number of decomposition mode components. $u_{k}(t)$ and $\omega_{k}$ denote the *k*-th mode and its center frequency, respectively. δ(t) is the Dirac delta function and * represents convolution. $\partial_{t}$ denotes the partial derivative with respect to time. $e^{-j\omega_{k}t}$ shifts the mode $u_{k}(t)$ to its baseband centered around $\omega_{k}$. The goal is to decompose the signal x(t) into a set of modes $u_{k}(t)$ that are each compact around their respective center frequencies $\omega_{k}$. By minimizing the sum of the squared norms of the modes’ analytic signals, VMD ensures that each mode is narrow-band and centered around a specific frequency.

Variational mode extraction (VME) is a specialized version of VMD that extracts a single mode with a predetermined center frequency. The basic principle behind VME is to decompose a signal into two parts: the desired mode $u_{d}(t)$ with the predetermined center frequency $\omega_{d}$ and the residual signal $u_{r}(t)$. To properly extract the desired mode, two conditions must be satisfied [33]: (1) the desired mode must be compacted around its center frequency; (2) the spectral overlap between the desired mode and the residual signal should be minimal. Furthermore, the original signal should be restricted by the summation of the extracted desired mode and the residual signal. The VME method can be described by the following constrained minimization problem:(2)$$\begin{array}{l} \underset{\left\{u_{d}(t)\},\{\omega_{d}\},\{u_{r}(t)\right\}}{\min}\left\{\alpha J_{1}+J_{2}\right\}\\ \begin{array}{ll} s.t. & u_{d}(t)+u_{r}(t)=x(t) \end{array} \end{array},$$
(3)$$J_{1}={\left\|\partial_{t}\left[\left(\delta (t)+\frac{j}{\pi t}\right)*u_{d}(t)\right]e^{-j\omega_{d}t}\right\|}_{2}^{2},$$
(4)$$J_{2}={\left\|\beta (t)*u_{r}(t)\right\|}_{2}^{2},$$
where β(t) denotes the impulse response of the filter, which is expressed as the following frequency response used to the residual signal:(5)$$\beta (\omega )=\frac{1}{\alpha {\left(\omega -\omega_{d}\right)}^{2}},$$
where α in Equations (2) and (5) is the penalty parameter for controlling the balance between $J_{1}$ (represents the compactness of the desired mode) and $J_{2}$ (represents the spectral overlap between the desired mode and residual signal).

Considering a quadratic penalty term and a Lagrange multiplier λ(t), Equation (2) can be modified as follows,
(6)$$\begin{array}{c} L(u_{d},\omega_{d},u_{r},\lambda )=\alpha {\left\|\partial_{t}\left[\left(\delta (t)+\frac{j}{\pi t}\right)*u_{d}(t)\right]e^{-j\omega_{d}t}\right\|}_{2}^{2}+{\left\|\beta (t)*u_{r}(t)\right\|}_{2}^{2}\\ \begin{array}{c} +{\left\|x(t)-\left(u_{d}(t)+u_{r}(t)\right)\right\|}_{2}^{2}+\langle \lambda (t),x(t)-\left(u_{d}(t)+u_{r}(t)\right)\rangle \end{array} \end{array}.$$

This minimization problem, like many VMD-based methods, may be addressed using the alternate direction method of multipliers algorithm (ADMM), which can convert a complex optimization problem [25,26,29,30]. The extracted mode, its corresponding center frequency, and the Lagrangian multiplier λ are constantly updated by the following Equations (7), (8) and (9), respectively, until the termination condition is fulfilled.
(7)$${\hat{u}}_{d}^{n+1}(\omega )=\frac{\hat{x}(\omega )+\alpha^{2}{\left(\omega -\omega_{d}^{n+1}\right)}^{4}{\hat{u}}_{d}^{n}(\omega )+\hat{\lambda }(\omega )/2}{\left[1+\alpha^{2}{\left(\omega -\omega_{d}^{n}\right)}^{4}\right]\left[1+2\alpha {\left(\omega -\omega_{d}^{n}\right)}^{2}\right]},$$
(8)$$\omega_{d}^{n+1}=\frac{\int_{0}^{\infty }\omega {\left|{\hat{u}}_{d}^{n+1}(\omega )\right|}^{2}d\omega }{\int_{0}^{\infty }{\left|{\hat{u}}_{d}^{n+1}(\omega )\right|}^{2}d\omega },$$
(9)$${\hat{\lambda }}^{n+1}={\hat{\lambda }}^{n}+\tau \left[\frac{\hat{x}(\omega )-{\hat{u}}_{d}^{n+1}(\omega )}{1+\alpha^{2}{\left(\omega -\omega_{d}^{n+1}\right)}^{4}}\right].$$

Finally, the above iteration terminates when the convergence condition is satisfied,
(10)$$\frac{{\left\|{\hat{u}}_{d}^{n+1}-{\hat{u}}_{d}^{n}\right\|}_{2}^{2}}{{\left\|{\hat{u}}_{d}^{n}\right\|}_{2}^{2}}<\varepsilon ,$$
where ${\hat{u}}_{d}^{n}(\omega )$ and $\omega_{d}^{n}$ represent the obtained desired mode and the center frequency in the *n*-th iteration, respectively. $\hat{x}(\omega )$ is Fourier transforms for the original signal. τ is one update parameter that is commonly preset to 0 to ensure the algorithm converges effectively. Another update parameter ε, which controls the reconstruction accuracy of the VME decomposition, is often set to a very tiny positive value (e.g., 1 × 10^−7^ in the following tests). The penalty parameter α and initial center frequency $\omega_{d}$ are important input parameters. Particularly, α determines the bandwidth of the desired mode, and a smaller value of α can yield a larger bandwidth.

### 2.2. Multivariate Variational Mode Extraction (MVME)

Multivariate variational mode decomposition (MVMD) is an extension of the standard variational mode decomposition (VMD) designed to handle multichannel or multivariate data. The goal of multivariate variational mode decomposition (MVMD) is to decompose a multivariate signal into a set of intrinsic mode functions (IMFs) that are common across all channels, ensuring that each mode has a narrow bandwidth and a specific center frequency [23]. Inspired by MVMD, multivariate VME (MVME) is proposed for extracting desired multivariate modulated oscillations $u_{d}(t),([u_{d,1}(t),u_{d,2}(t),\ldots ,u_{d,C}(t)])$ from input data $x(t),([x_{1}(t),x_{2}(t),\ldots ,x_{C}(t)])$ containing *C* number of data channels. The constrained optimization problem for MVME can be described according to Equation (2) as:(11)$$\begin{array}{l} \underset{\left\{u_{d}(t)\},\{\omega_{d}\},\{u_{r}(t)\right\}}{\min}\left\{\alpha J_{1}+J_{2}\right\}\\ \begin{array}{ll} s.t. & u_{d}(t)+u_{r}(t)=x(t) \end{array} \end{array},$$
(12)$$J_{1}={\left\|\partial_{t}\left[\left(\delta (t)+\frac{j}{\pi t}\right)*u_{d}(t)\right]e^{-j\omega_{d}t}\right\|}_{2}^{2},$$
(13)$$J_{2}={\left\|\beta (t)*u_{r}(t)\right\|}_{2}^{2}.$$

Considering a quadratic penalty term and a Lagrange multiplier, Equation (11) can be modified as follows,
(14)$$\begin{array}{l} L(u_{d},\omega_{d},u_{r},\lambda )=\alpha J_{1}+J_{2}\\ \begin{array}{c} \;+{\left\|x(t)-\left(u_{d}(t)+u_{r}(t)\right)\right\|}_{2}^{2} \end{array}+\langle \lambda (t),x(t)-\left(u_{d}(t)+u_{r}(t)\right)\rangle \end{array}$$

The extracted mode and its corresponding center frequency are constantly updated by Equations (16) and (17), respectively, until the termination condition is fulfilled.
(15)$${\hat{u}}_{d,c}^{n+1}(\omega )=\frac{{\hat{x}}_{c}(\omega )+\alpha^{2}{\left(\omega -\omega_{d}^{n+1}\right)}^{4}{\hat{u}}_{d,c}^{n}(\omega )+\frac{{\hat{\lambda }}_{c}^{n}(\omega )}{2}}{\left[1+\alpha^{2}{\left(\omega -\omega_{d}^{n+1}\right)}^{4}\right]\left[\left[1+2\alpha {\left(\omega -\omega_{d}^{n}\right)}^{2}\right]\right]},$$
(16)$$\omega_{d}^{n+1}=\frac{\sum_{c}\int_{0}^{\infty }\omega {\left|{\hat{u}}_{d,c}^{n+1}(\omega )\right|}^{2}d\omega }{\sum_{c}\int_{0}^{\infty }{\left|{\hat{u}}_{d,c}^{n+1}(\omega )\right|}^{2}d\omega }.$$

Finally, the above iteration terminates when the convergence condition is satisfied,
(17)$$\frac{{\left\|{\hat{u}}_{d,c}^{n+1}-{\hat{u}}_{d,c}^{n}\right\|}_{2}^{2}}{{\left\|{\hat{u}}_{d,c}^{n}\right\|}_{2}^{2}}<\varepsilon ,$$
where *c* is the channel number of data.

### 2.3. The Proposed Interference Elimination Method

In the application of seismic interference elimination, the desired modes are the interferences produced by the mechanical vibrations or power facilities. As mentioned, the initial center frequencies and the desired modes’ penalty parameters have a significant impact on the extraction effect of MVME. The fundamental goal of the proposed interference elimination method is to identify and mitigate unwanted noise or signals that may obscure or distort the effective seismic signal. It is important to maximize the separation of effective seismic signals from interference noise to ensure that the desired mode contains complete and relatively pure interference through adopting an appropriate initial value of ω_d_ and penalty parameter α. The interference produced by mechanical vibrations or power facilities often has stable dominant frequencies that last for a long time. Therefore, we first identify the frequency components that need to be eliminated from the recordings, and then adjust the penalty parameters to achieve the optimal de-noising result.

To identify the frequency components in time sequence recording, frequency-based and time-frequency-based methods are commonly used, such as the Fourier transform (FT), autoregressive (AR) modeling, short-time Fourier transform (STFT), and wavelet transform (WT). In this study, the value of ω_d_ was selected using synchrosqueezing-based time-frequency analysis (SST). This advanced method enhances the resolution of time-frequency representations by building on traditional transforms like the STFT or WT. It reassigns the time-frequency coefficients to sharpen the representation, producing a more precise and concentrated time-frequency representation, making it easier to identify and interpret signal components [38,39,40]. The S/N of the target seismic signal after removing interferences is used to favor the optimal value α from a range of values. The S/N is calculated using the following equation,
(18)$$S/N=20\log_{10}\left(\frac{A_{s}}{A_{n}}\right),$$
where *A*_s_ and *A*_n_ represent the root-mean-square amplitudes in a time window of seismic signals and background noise, respectively.

The process of the proposed method is summarized as follows:(1)Determine the number (*M)* and the frequencies (*w*_1_, *w*_2_, …, *w*_M_, sorted by energy level) of pending strong interferences according to the seismic recordings X(t);(2)Set m = 1, the range and step of the penalty parameter (α_min_, α_max_, Δα), and define X’(t) = X(t);(3)Decompose the seismic data using MVME with the initial frequency *w_m_* and a series of penalty parameters. Calculate the S/Ns of seismic signals in the residual part and determine the optimal value α_m_ of penalty parameter according to the maximum of S/Ns.(4)Extract the *m*-th interference noise from seismic data using MVME with the optimal penalty parameter α_m_. Define the *m*-th interference as **u**_m_(t) (**u**_m_(t) = [*u*_m,1_(*t*), … *u*_m,c_(*t*), …, *u*_m,C_(*t*)]; c represents the channel. The residual signal after removing *m*-th interference can be obtained by X’(t) = X(t) − **u**_m_(*t*);(5)If m < M, increment m by 1 and repeat steps (3) and (4); Otherwise, go to step (6).(6)Finally, X’(t) is the de-noised seismic record which reduces the interferences.

Figure 1 illustrates the flowchart of the proposed interference elimination method.

## 3. Numerical Analysis

In this section, we evaluated the performance of the proposed method using synthetic data. The synthetic data were composed of a Ricker wavelet (regarded as the seismic signal) with the peak frequency of 50 Hz (Figure 2a), a non-stationary signal (regarded as the interference noise) (Figure 2b), and Gaussian white noise **n**(*t*) with a variance of 0.1:(19)$$\begin{array}{l} x(t)=A\left[\begin{array}{c} signal_{3}(t)\\ signal_{4}(t) \end{array}\right]+n(t)\\ signal_{3}(t)=(1-2{\left(\pi f(t-t_{0})\right)}^{2})e^{-{\left(\pi f(t-t_{0})\right)}^{2}},t_{0}=500\\ signal_{4}(t)=(1+0.4sin(5\pi t))*cos(2\pi 25t+6\pi t^{2}) \end{array}.$$

The amplitude ratios of the seismic signal and interference noise on the 3-C record were set to 1:2:3 and 5:4:3 in mixing matrix **A**, respectively.

We investigated the effects of the penalty parameter and initial center frequency on MVME performance. From the time-frequency analysis result (as shown in Figure 2d), it is evident that there was a strong interference in the record with a dominant frequency of about 26 Hz. To find the optimal α value, we fixed the initial center frequency at 26Hz and initialized α with a value of 2000, increasing it in steps of 2000 up to 200,000. We extracted and removed the interference signals using MVME with the predefined center frequencies. The optimal penalty parameter was determined by maximizing the S/N of the de-noised signals within a window of 40 samples following seismic arrival. When the penalty parameter was set to 12,400, the de-noised results were optimal, as shown in Figure 3. Theoretically, α determines the bandwidth of the desired mode; a smaller α yields a larger bandwidth, while a larger α yields a smaller bandwidth. The processing results using different penalty parameters are shown in Figure 4, in which small α indicates underprocessing and large α indicates overprocessing.

In addition, we compared the effect of different initial center frequencies on the extraction results by fixing α = 124,000. With ω_d_ ranging from 15 to 30 Hz, we calculated and compared the S/Ns of the de-noised signals and the final center frequency after iteration. As long as the ω_d_ value is estimated within an appropriate range, MVME exhibits consistent performance for interference extraction. The results, shown in Figure 5, indicate that the initial center frequency had a negligible effect on the results. In practice, the time-frequency-based method can help identify the range of ω_d_, allowing for easy determination of an acceptable or optimal value of ω_d_.

We also processed each of the three components using the VME method with identical parameters and the optimal α value. The processing results are presented in Figure 6a,b. Unlike the extraction results of components 2 and 3, signal component 1 showed a significant discrepancy compared to the actual signal, leading the underprocessing for the seismic wavelet signal. This discrepancy arose from variations in the energy distribution of the signal, influenced by differences in optimization outcomes. A comparison with the results shown in Figure 4c,d indicates that the MVME method was more effective at achieving synchronous analysis of multitrace data than the VME method.

For comparison, we also used the conventional notch filters and synchrosqueezing transform method (SST) to eliminate the interferences. In the SST method, we identified ridges in the magnitudes of the synchrosqueezed transform and reconstructed them along the ridge to isolate and analyze modes. Figure 7a,c shows the removed interferences and de-noised signals using the notch filter, while Figure 7b,d presents the results using the SST method. It is evident that the notch filter failed to completely eliminate the interferences. Specifically, we noted that while the SST method was effective in separating seismic wavelet signals from interferences, it altered the shape and amplitude of the wavelet signal, as observed in Figure 7c,d. Although the de-noised waveforms obtained using the SST method appear superior to those obtained with the MVME method (Figure 4c,d), the SST method was deficient in preserving the amplitude of the seismic wavelet signals compared to the MVME method.

Quantitative evaluation of amplitude-preserving de-noising is crucial for assessing the performance of different methods. In this study, we used different absolute amplitudes for the seismic signals in this example and generated a series of synthetic 3-C records with the S/Ns ranging from −0.17 dB to 8.27 dB (as shown in Figure 8). We calculated the S/Ns after interference elimination and the root mean square (RMS) errors between the de-noised signals and noise-free data to demonstrate the effects of the above three methods on preserving the seismic signal. We calculated the RMS errors using the following equation:(20)$$RMS=\sqrt{\frac{1}{N}\sum_{i=1}^{N}{\left(X_{clean}(i)-X_{denoised}(i)\right)}^{2}},$$
where X_clean_ and X_denoised_ are the free-noise recording and de-noised recording, respectively. N is the number of samples in the modeled recording. Additionally, we calculated the polarization parameters of the 3-C seismic signal using the following equations,
(21)$$\left\{\begin{array}{l} L=\frac{{\left(\lambda_{1}-\lambda_{2}\right)}^{2}+{\left(\lambda_{2}-\lambda_{3}\right)}^{2}+{\left(\lambda_{1}-\lambda_{3}\right)}^{2}}{2{\left(\lambda_{1}+\lambda_{2}+\lambda_{3}\right)}^{2}}\\ \theta =\arctan\left(\frac{e_{1y}}{e_{1x}}\right)\\ \phi =\arccos\left(e_{1z}\right) \end{array}\right.,$$
where *L* is the rectilinearity, θ and ϕ are the polarization angles (θ represents the azimuth angle and ϕ represents the inclination angle). $\lambda_{1},\lambda_{2},\lambda_{3}(\lambda_{1}>\lambda_{2}>\lambda_{3})$ are the eigenvalues of a covariance matrix constructed using windowed waveforms at seismic arrival, and $\left[e_{1x},e_{1y},e_{1z}\right]$ is the eigenvector corresponding to the largest eigenvalue $\lambda_{1}$. These polarization parameters help in understanding the direction of the seismic wave propagation, which is crucial for accurate seismic/microseismic data interpretation.

Figure 8 show the comparison of the processing results of three methods for synthetic data with varying S/Ns. The notch filter failed to completely remove the interferences and performed poorly for recordings with low S/Ns. Since synthetic 3-C records only changed the amplitude of the seismic wavelet signal without changing the amplitude of the interferences, the RMS error after notch filter de-noising at different S/Ns did not vary significantly (Figure 8b), being primarily affected by random noise. Both the SST method and the MVME exhibited superior performance in terms of rectilinearities, with values close to 1. Although the SST method achieved a high S/N, it resulted in a loss of effective seismic signal, as evidenced by the RMS error. In addition, this method affected the rectilinearities and polarization angles of the three-component seismic signals. The proposed method, on the other hand, enhanced the S/N with a reduced RMS error of the effective seismic signal. Furthermore, the polarization characteristics after removing interference closely matched the actual values (represented by black dotted lines). These observations indicate that the proposed method can effectively remove the strong interferences while preserving the seismic signals.

## 4. Field Data Test

In this section, we applied the proposed method to two field datasets. The field data underwent a preliminary processing phase, which included receiver orientation and band-pass filtering. The selection of frequency range for band-pass filtering was tailored to align with the specific characteristics of the field data to eliminate the low-frequency and high-frequency noises. By focusing on the [5, 200] Hz range, we ensured that the filtering process targeted the frequencies carrying the most relevant information for our analysis. After removing the mechanical vibration interferences, we further calculated and compared the S/Ns and the rectilinearities of the remaining signals to demonstrate the effectiveness of the proposed method.

### 4.1. Active Seismic Data Example

The first field dataset was acquired from a seismic monitoring survey in a heavy oil field steam-assisted gravity drainage (SAGD) project in Xinjiang, China. A network of 200 buried 3-C receivers installed in shallow wells with a depth of about 10m and a sampling interval of 1 ms was utilized for monitoring. The acquisition system was located above the horizontal SAGD wells, close to operational oil pumps. The target reservoir area was covered by 2668 active sources for seismic reflection imaging. Figure 9 shows the survey geometry in this case. Figure 10a shows the waveforms of an active shot recorded by three adjacent receivers, in which the red line indicates the manually picked direct P-wave arrival time. It shows that there was a strong interference noise with the dominant frequency of about 15 Hz (as shown in Figure 10b, especially in the horizontal components) in these data. Based on the distance of the receivers relative to the wellheads, it was confirmed that this low-frequency interference was generated by the nearby operational oil pump.

We applied the proposed method to reduce the strong interference noise, the initial center frequency ω_d_ was set to 15, and the penalty parameter α varies from 1000 to 100,000 with an interval of 1000. In this case, the optimal penalty parameter was calculated by taking the maximum value of S/N of the de-noised signals in the window following seismic arrivals. Considering that the target reservoir was located at a depth of 300 m, a 6 s time window was selected to retain as much reflected wave information as possible. The optimal penalty parameter was determined by maximizing the S/N of the de-noised signals. When the penalty parameter was set to 29,000, the de-noised results were optimal. Waveforms in Figure 11a,b show the de-noised signals and the removed interferences via the proposed method. It can be seen from the de-noised waveforms that seismic signals were much clearer, particularly in the horizontal components. To quantitatively analyze the effectiveness of the de-noising, we calculated the S/Ns and rectilinearities of the direct P-wave in 3-C seismic signals. The time window used for these calculations encompassed 50 sampling numbers after the arrival of the direct P-wave. The results are listed in Table 1 and Table 2.

For comparison, the notch filter and SST method were also used to eliminate the interferences. Waveforms in Figure 12a,b show the de-noised signals and the removed interferences via the notch filter, while waveforms in Figure 12c,d show the results obtained using the SST method. Both methods were capable of suppressing the interference noise, as evidenced by the increased S/Ns (Table 1) and the improved rectilinearities of the direct P-waves in the de-noised signals (Table 2). While the de-noising results of the three methods were acceptable in different scenarios, a closer examination of the de-noised waveforms revealed distinct differences. The results from the notch filter still contained residual interferences, indicating underprocessing (as shown in Figure 12a). Additionally, the SST method tended to overprocess, removing some effective seismic signals with frequencies close to the interference. Furthermore, the processing results show varying de-noising degrees across the 3C records, such as the 1Z and 2Z components in Figure 12c. By comparison, the proposed method achieved superior results in suppressing the interferences and preserving seismic signals. It effectively removed the strong interferences while minimizing the loss of useful seismic information, thus providing a more balanced and accurate de-noising solution.

### 4.2. Microseismic Data Example

The second field dataset was recorded from microseismic monitoring of a hydraulic fracturing job in a shale gas play in Chongqing, China. A temporary string composed of 14 levels of triaxle 15-Hz geophones was deployed in the inclined section of a horizontal well to monitor the stimulations. The data sampling interval was 0.25 ms, and the receiver spacing interval was 10 m. Figure 13 shows the survey geometry in this case. Since the wellhead of the monitoring well was close to that of the treatment well, the downhole geophones continuously recorded fracturing pump vibration signals.

Figure 14a shows 3-C waveforms of one microseismic event recorded on the shallowest receiver. It is obvious that the arrivals of direct P- and S-waves were masked by strong interferences. Via time-frequency analysis of N-component data, the dominant frequencies of the interference noises in the microseismic recordings were determined. The three dominating frequencies (60 Hz, 110 Hz and 20 Hz, ranked from strongest to weakest) of the monitoring data are displayed in the time-frequency representation in Figure 14b.

We extracted and removed the above three interference frequencies using the proposed method. Waveforms in Figure 15a,b show the de-noised signals and the removed interferences via MVME. The arrival of the microseismic signals were more apparent after removal of interference noise. The time windows for calculating the signal-to-noise ratio (S/N) and rectilinearity are defined as follows: for the P-wave, the time window was 100 sampling numbers after the P-wave arrival; for the S-wave, the time window was 200 sampling numbers after the S-wave arrival. The S/N calculation time window for the entire microseismic event was 2000 sampling numbers after the P-wave arrival. We can see from Table 3 and Table 4 that the S/Ns and the rectilinearities of the microseismic signals and P-wave were also increased using the proposed method. We also applied the notch filters and SST method to the same dataset for comparison. The de-noised signals and the removed interferences obtained using the notch filters are shown in Figure 15c,d, and the results obtained using the SST method are shown in Figure 15e,f. Both methods struggled to eliminate the interferences, which was reflected in the S/Ns (as shown in Table 3) and rectilinearities of the P-wave (as shown in Table 4). Although the notch filter was applied with carefully selected parameters, it failed to eliminate interferences in two horizontal components. The de-noised signals indicated that the notch filter resulted in insufficient processing and had limited improvement in the S/N of the microseismic event. Although the SST method significantly enhanced the S/Ns, the processing results show that the amplitudes of the P-waves were corrupted (as shown in Figure 15f). In comparison, the proposed method demonstrated more pronounced advantages in de-noising effectiveness and preserving the effective P-wave signal. The actual data contained weak effective S-wave signals that overlapped significantly in frequency with interference signals and it poses a challenging problem for current de-noising methods. The methods mentioned in this paper showed no significant improvement in de-noising under such conditions. However, after applying the proposed method, the S-wave arrival became more apparent, aiding in accurate source localization in subsequent processing.

## 5. Conclusions

Long-duration strong interferences caused by mechanical vibrations or power facilities during active and passive seismic surveys contaminate the seismic signals and must be removed from the raw data. In this paper, we presented an interference noise suppression method for multitrace seismic/microseismic data. The proposed method employs MVME to extract and separate individual modes from seismic records based on the approximate center frequencies and optimal penalty parameters. Testing of the proposed method on synthetic data have demonstrated its effectiveness in removing the mechanical vibration interference noise. In addition, we found that a suitable range of initial center frequencies had nearly no impact on the interference extraction via the proposed method. The penalty parameters, which mainly affect the complete removal of interference noise, can be optimized based on the S/N of the seismic signals after de-noising. By applying the proposed method to field datasets, we have shown that the proposed method performed better than conventional methods in suppressing the interferences and preserving seismic signals.

## Figures and Tables

**Figure 1 sensors-24-07399-f001:**
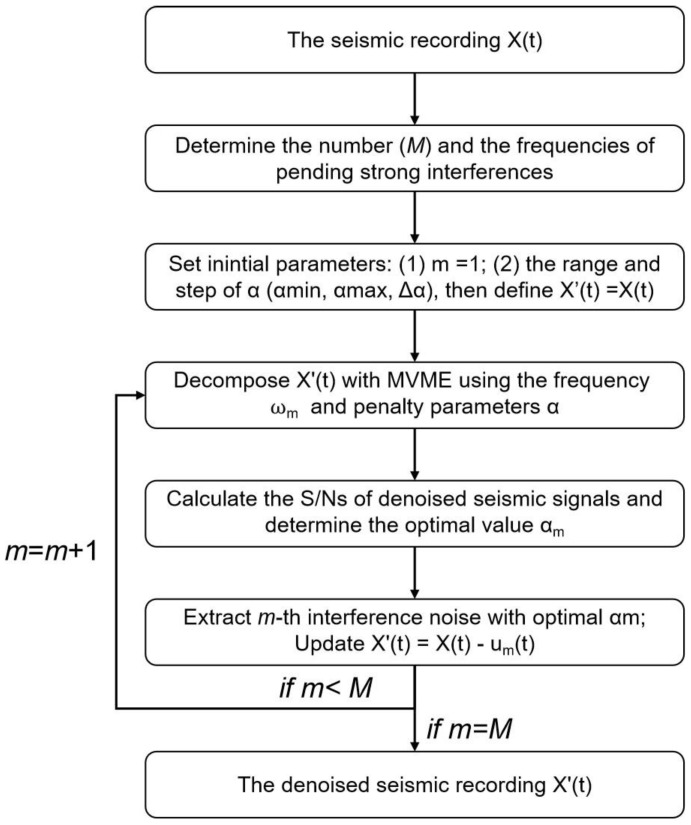
Flowchart of the proposed method.

**Figure 2 sensors-24-07399-f002:**
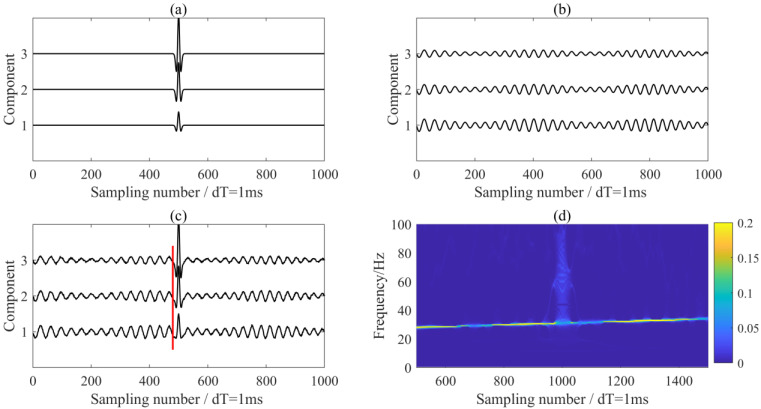
Waveforms of (**a**) the 3-C seismic signal, (**b**) the 3-C interference noise, and (**c**) the synthetic data. (**d**) The time-frequency analysis result of the 1st component data. The red lines in (**c**) indicate the seismic arrival.

**Figure 3 sensors-24-07399-f003:**
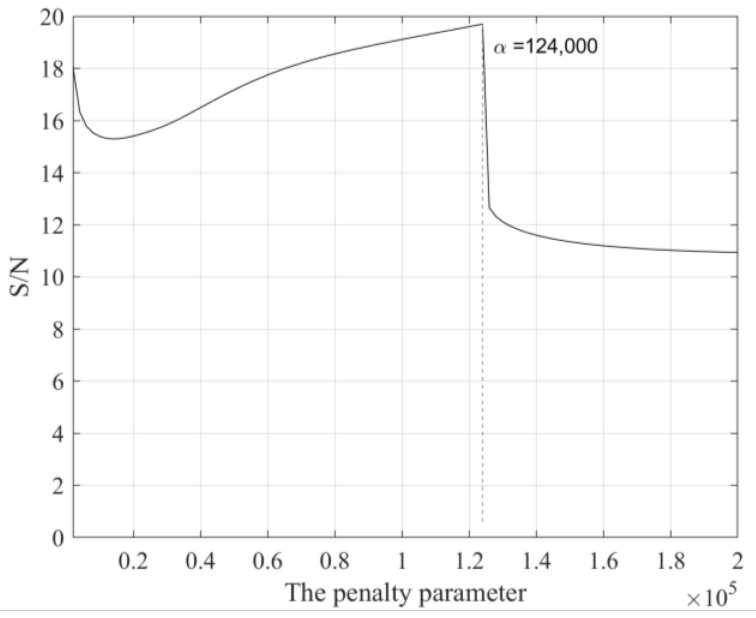
The S/Ns of the residual signal after removing interference using MVME with a series of penalty parameters. Selection of optimal parameters according to the maximum value of S/N.

**Figure 4 sensors-24-07399-f004:**
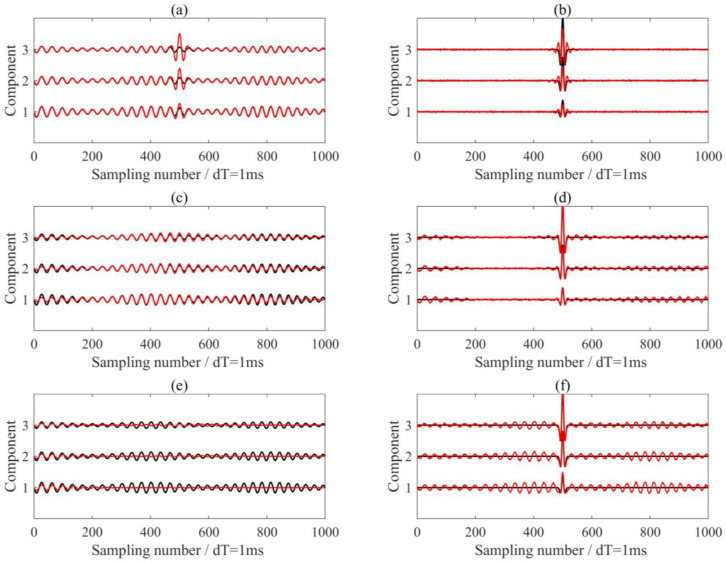
Comparison of the processing results using MVME with different penalty parameters α. (**a**,**c**,**e**) These show the extracted interference noise using α = 2000, 124,000, 200,000, respectively. (**b**,**d**,**f**) These show the de-noised signals using α = 2000, 124,000, 200,000, respectively. The black and red lines are the real signals in the synthetic data and the processing results, respectively.

**Figure 5 sensors-24-07399-f005:**
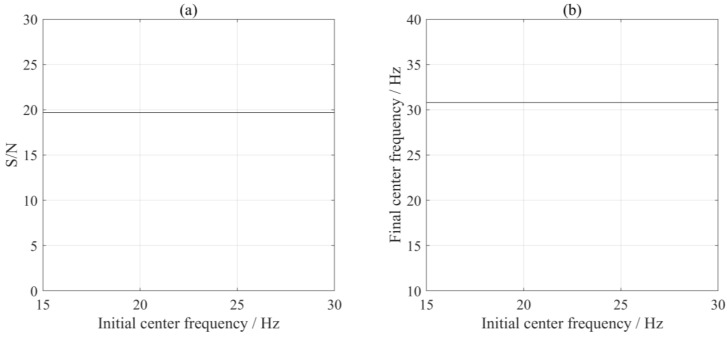
Comparison of the S/Ns (**a**) and the final center frequencies (**b**) using MVME with different initial center frequencies.

**Figure 6 sensors-24-07399-f006:**
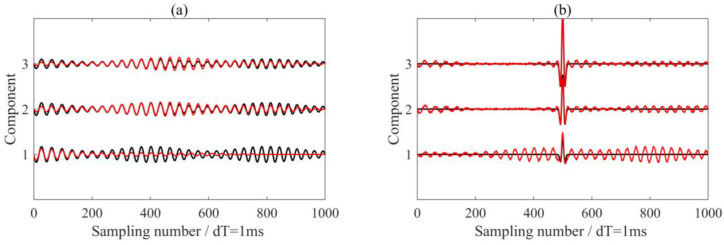
The processing results using the VME method with the optimal penalty parameter α = 124,000. (**a**) The extracted interference noise. (**b**) The de-noised signals. The black and red lines are the real signals in the synthetic data and the processing results via the VME method, respectively.

**Figure 7 sensors-24-07399-f007:**
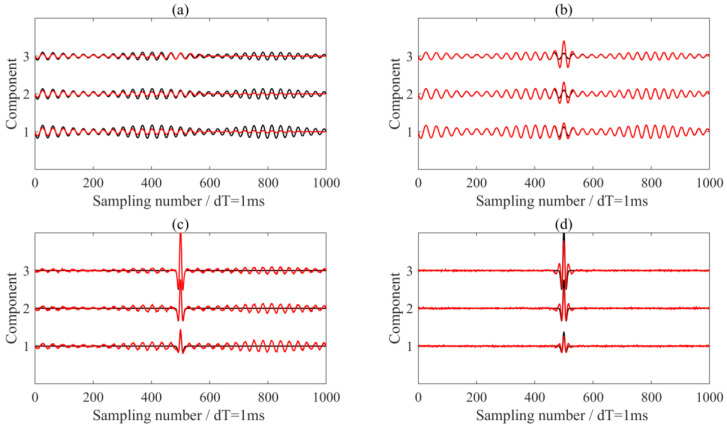
The processing results using the notch filter and the SST method. (**a**,**b**) The removed interferences via notch filter and SST method, respectively. (**c**,**d**) The de-noised signals via notch filter and SST method, respectively. The black and red lines are the real signals in the synthetic data and the processing results, respectively.

**Figure 8 sensors-24-07399-f008:**
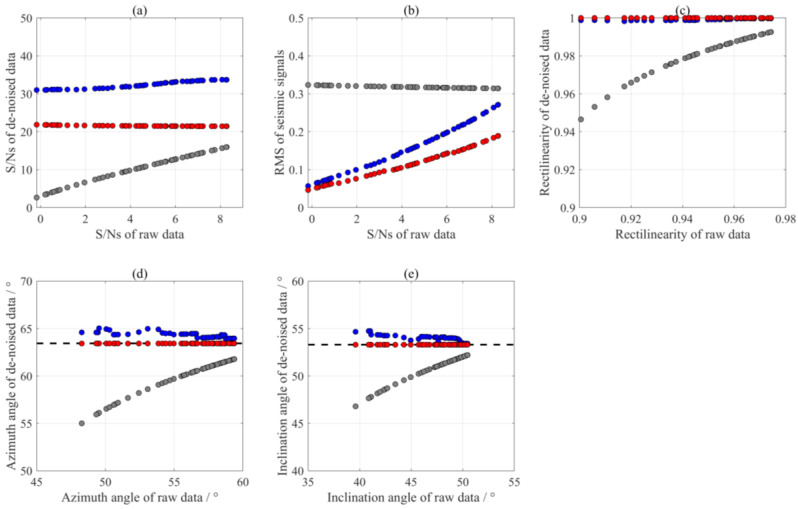
Comparison of the processing results of notch filter (grey dots), SST method (blue dots), and the proposed method (red dots). (**a**) S/Ns; (**b**) RMS; (**c**) rectilinearity; (**d**) azimuth angle; and (**e**) inclination angle.

**Figure 9 sensors-24-07399-f009:**
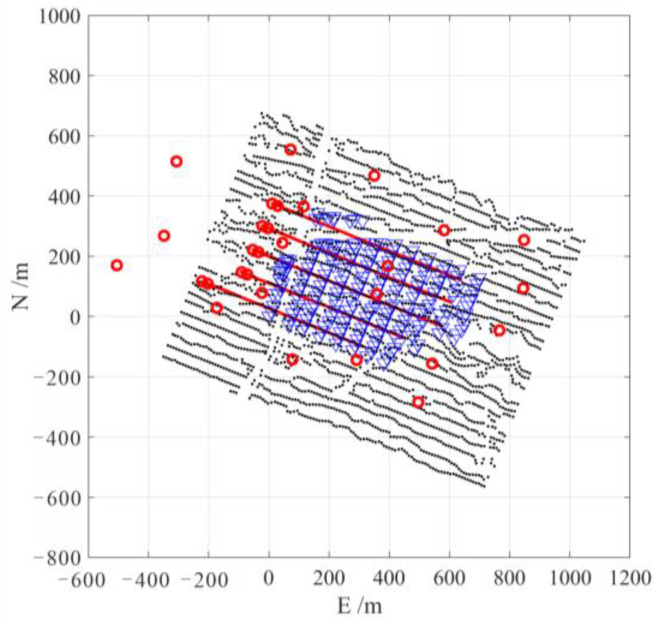
The planar view of the active seismic survey geometry. Black dots, blue triangles, and red circles represent the location of active seismic shots, 3-C receivers, and wellheads, respectively. The red lines indicate the horizontal well trajectories.

**Figure 10 sensors-24-07399-f010:**
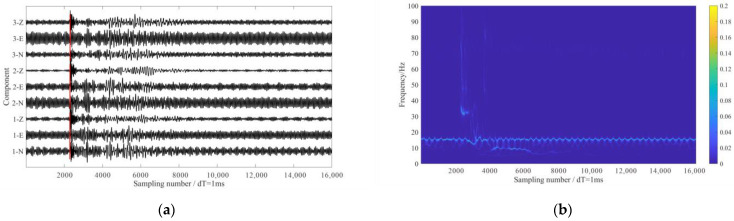
The 3-C seismic data from a SAGD seismic monitoring project. (**a**) The 3-C waveforms of an active shot. (**b**) The time-frequency analysis result of 1-N component data. The red lines in (**a**) indicate the manually picked direct P-wave arrival times.

**Figure 11 sensors-24-07399-f011:**
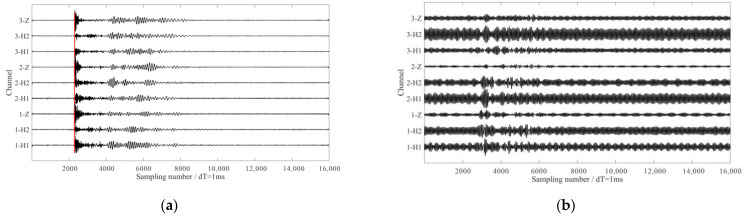
The processed results using the proposed method. (**a**,**b**) The de-noised signals and the removed interferences, respectively. The amplitude ranges of waveforms are the same. The red lines in (**a**) indicate the manually picked direct P-wave arrival times.

**Figure 12 sensors-24-07399-f012:**
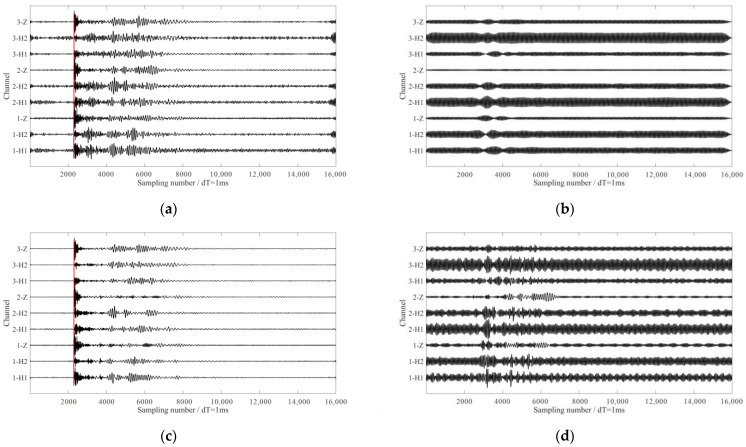
The processing results using the notch filter and SST method. (**a**,**b**) The de-noised signals and the removed interferences via notch filter, respectively. (**c**,**d**) The de-noised signals and the removed interferences via the SST method, respectively. The red lines in (**a**,**c**) indicate the manually picked direct P-wave arrival times.

**Figure 13 sensors-24-07399-f013:**
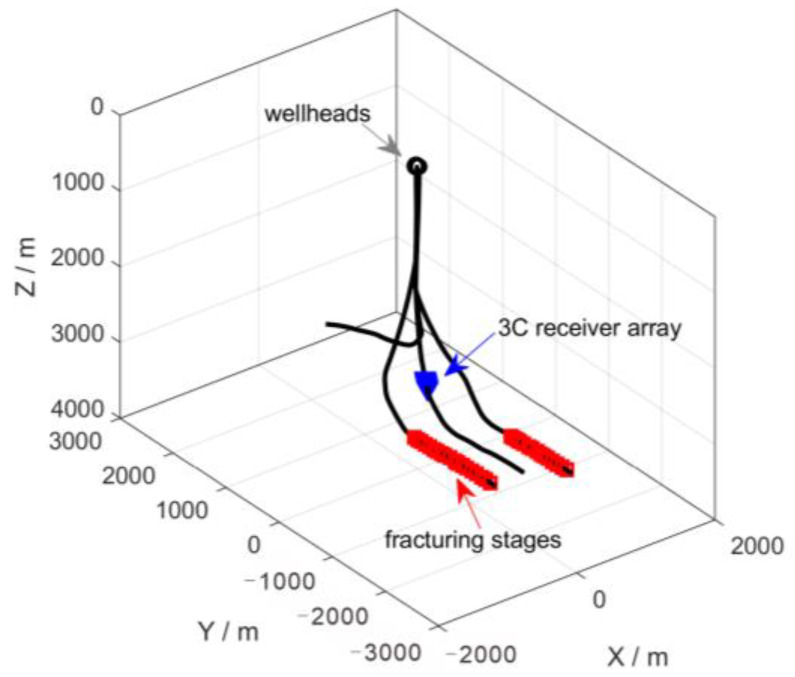
The 3D view of the survey geometry. The wellhead of the monitoring well was near to that of the treatment.

**Figure 14 sensors-24-07399-f014:**
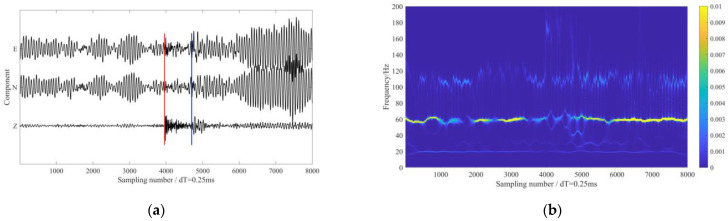
The 3-C microseismic data from a hydraulic fracturing downhole monitoring. (**a**) The 3-C waveforms of a microseismic event. (**b**) The time-frequency analysis of N-component data. The red and blue lines in (**a**) represent the manually picked P-wave and S-wave arrival times, respectively.

**Figure 15 sensors-24-07399-f015:**
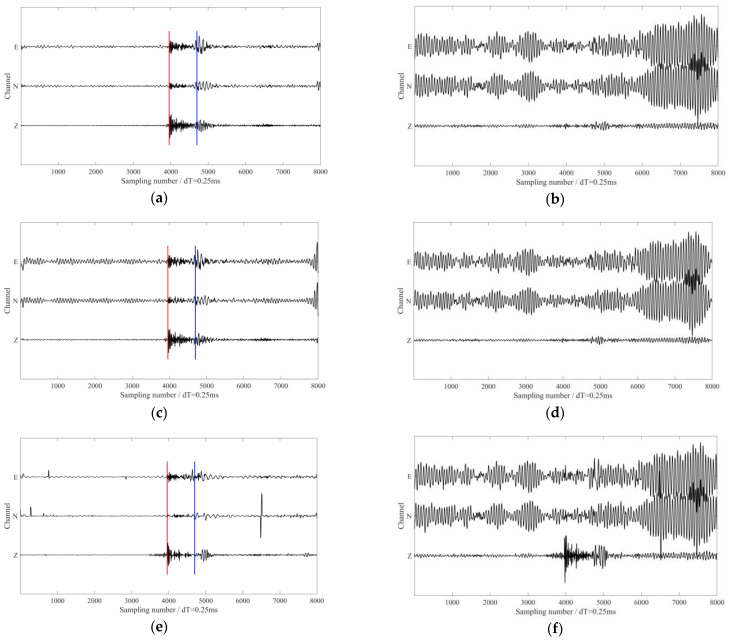
The processing results using different methods. (**a**,**b**) The de-noised signals and the removed interferences via the proposed methods, respectively. (**c**,**d**) The de-noised signals and the removed interferences via notch filter, respectively. (**e**,**f**) The de-noised signals and the removed interferences via the SST method, respectively. The red and blue lines in (**a**,**c**,**e**) represent the manually picked P-wave and S-wave arrival times, respectively.

**Table 1 sensors-24-07399-t001:** Comparison of the S/Ns (dB) of the direct P-wave in the de-noised seismic signals using different methods.

Receiver No.	Raw Records	Notch Filter	SST	The Proposed Method
1	9.80	16.52	23.33	23.60
2	6.52	16.67	24.72	24.40
3	5.56	18.96	25.50	26.58

**Table 2 sensors-24-07399-t002:** Comparison of the rectilinearities of the direct P-wave in the de-noised seismic signals using different methods.

Receiver No.	Raw Records	Notch Filter	SST	The Proposed Method
1	0.47	0.85	0.88	0.89
2	0.13	0.92	0.94	0.95
3	0.24	0.95	0.99	0.98

**Table 3 sensors-24-07399-t003:** Comparison of the S/Ns (dB) of the de-noised microseismic signals using different methods.

	Raw Records	Notch Filter	SST	The Proposed Method
Microseismic event	−0.46	6.58	12.49	13.39
P-wave	5.10	13.51	11.58	15.38
S-wave	5.80	6.26	0.75	6.61

**Table 4 sensors-24-07399-t004:** Comparison of the rectilinearities of the de-noised microseismic signals using different methods.

	Raw Records	Notch Filter	SST	The Proposed Method
P-wave	0.52	0.82	0.64	0.81
S-wave	0.28	0.27	0.18	0.27

## Data Availability

The data and code presented in this study are available on request from Z.Y.

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
