# Peer review of "Strong Interference Elimination in Seismic Data Using Multivariate Variational Mode Extraction"

_sensors, 2024, doi:10.3390/s24227399_

Round 1
Reviewer 1 Report
Comments and Suggestions for Authors
1、This manuscript proposes a seismic data denoising method using the multivariate variational mode extraction (MVME). The MVME is designed to handle multichannel or multivariate data. It is an extension of the variational mode extraction (VME). Please add the advantages of MVME method over VME method in seismic data denoising.
2、In the introduction section, authors summarized the existing methods for seismic noise suppression, including the frequency-domain filters, morphological component analysis, and time-frequency analysis methods, et al. In addition to these methods mentioned by authors, there is also an important class of algorithms, sparse representation based denoising methods.
Turquais, et al., A method of combining coherence-constrained sparse coding and
dictionary learning for denoising. Geophysics, 2017, 82(3), V137-V148.
Shao, et al., Simultaneous denoising of multicomponent microseismic data by joint sparse representation with dictionary learning. Geophysics, 2019, 84(5), KS155-KS172.
3、The advanced denoising methods such as unsupervised deep learning are mentioned in the introduction section. The authors conclude that the drawback of unsupervised deep learning methods is the need for high-quality labeled data. Supervised deep learning methods require labeled data, while unsupervised learning methods do not require labeled data.
Shao, et al., Siamese network based noise elimination of artificial seismic data recorded by distributed fiber-optic acoustic sensing. Chinese J. Geophys, 2022, 65(9): 3599-3609.
4、In the field data example, how is the SNR calculated in case of unknown clean data?
Comments on the Quality of English Language
The English language requires minor editing.
Reviewer 2 Report
Comments and Suggestions for Authors
Only after a major revision the article can be published in the journal Sensors.

Reviewer 3 Report
Comments and Suggestions for Authors
1. Please add punctuation after the formulas in the article. For example, there is no punctuation after formula (1), and the format of the following line of text is incorrect. This issue occurs multiple times throughout the article, so please make uniform modifications.
2. Lines 226-238 primarily describe the workflow of the article; however, there is no mention of the settings and usage of the parameters ω_dand α in this section. The introduction to the workflow lacks detail, and I recommend adding a flowchart to enhance the reader's understanding.
3. The image resolution is insufficient, and the font size and style of the axis labels are not standardized. Please provide the original uncompressed image.
4. In lines 286-293, the testing of the wd data does not use the optimal value of α, which is 11800, but instead uses α=24000. Please clarify the reasoning behind this choice.
5. Consider adding the calculation formula for RMS.
6. The title of Figure 5 is not sufficiently clear, and the formatting is also unclear.
7. In the "Numerical Analysis" section, to test the value of α, it is also necessary to test the ω_d value (using the SST method) and compare the advantages of the MVME method against the other two methods. Please reorganize this content with detailed divisions to make it easier for readers to understand.
Some of the sentences seem hard to understand, I suggest that it is edited by native speakers.
Reviewer 4 Report
Comments and Suggestions for Authors
A method for eliminating mechanical vibration interferences in seismic data was propose by authors. Such a method, is an extension of the variational mode extraction (VME) technique to a multivariate form. The authors are interested removing interference frequencies from seismic data using the proposed method MVME with optimal balancing factors.
The authors perform a comparison with notch filter and time-frequency transform method. we applied the proposed method to two field datasets. They used field data pre-processed by receiver orientation and band-pass filtering with a passband of [5, 200]Hz .
There is a reason to eliminate low and high frequencies?
After removing the mechanical vibration interferences, we further calculated and compared the S/Ns and the rectilinearities of the remaining signals to demonstrate the effectiveness of the proposed method.
The proposed method was applied to reduce the strong interference noise showed in Figure 9, however, the plots indicate the data set of three received.
These received correspond with the position showed in the Figure 12 arrow?
The results are summarized in the figure 14 and Tables 3 and 4 which describe the differences observed between the method proposed with the other two. In Table 4 the values for the three methods are similar when the analysis was performed for S wave.
The authors must explain this behavior and specify the limits of their method.
Round 2
Reviewer 2 Report
Comments and Suggestions for Authors
In response to the comments and the new version of the article, the authors clarified all the unclear points that I found in the old version of the article.
The reviewed article is very interesting and has undoubted general scientific and practical importance in the field of processing of seismic signals. This version of the article may be published in the journal Sensors.
Author Response
Thank you for your positive feedback and for recognizing the improvements in the revised version of our article. We are pleased to hear that the clarifications addressed all previously unclear points and that the article's scientific and practical significance in the field of seismic signal processing is evident.